# Using Myofascial Therapy to Improve Psychological Outcomes, Quality of Life, and Sexual Function in Women with Chronic Pelvic Pain—A Case Series

**DOI:** 10.3390/healthcare12030304

**Published:** 2024-01-24

**Authors:** Esther Diaz-Mohedo, Fidel Hita-Contreras, Eduardo Castro-Martin, Andrzej Pilat, Borja Perez-Dominguez, Geraldine Valenza-Peña

**Affiliations:** 1Department of Physiotherapy, University of Malaga, 29071 Malaga, Spain; estherdiaz@uma.es; 2Department of Health Sciences, Faculty of Health Sciences, University of Jaen, 23071 Jaen, Spain; fhita@ujaen.es; 3Department of Physiotherapy, Faculty of Health Sciences, University of Granada, 18016 Granada, Spain; eduardoc@ugr.es (E.C.-M.); valenzagera@hotmail.com (G.V.-P.); 4Myofascial Therapy School “Tupimek”, Physiotherapy School ONCE, Universidad Autonoma, 28049 Madrid, Spain; apilat@tupimek.com; 5Department of Physiotherapy, University of Valencia, 46010 Valencia, Spain

**Keywords:** musculoskeletal manipulations, myofascial pain syndromes, pelvic pain, quality of life, sexual dysfunction, physiological

## Abstract

(1) Background: Chronic pelvic pain represents a prevalent condition afflicting women. Research has highlighted the presence of psychological distress and sexual dysfunction in these individuals. Regrettably, myofascial pelvic pain often goes unnoticed and untreated despite its integral role in chronic pelvic pain. (2) Methods: By employing a longitudinal case series design, we studied eighteen women afflicted with chronic pelvic pain. Over a 12-week period, these participants underwent 15 sessions of myofascial therapy. Data encompassing sociodemographic particulars, the Hospital Anxiety and Depression Scale, the Medical Outcomes Study Short Form 12 questionnaire, and the Female Sexual Function Index were collected at baseline, 12 weeks post-intervention, and again at the 24-week mark. (3) Results: After a span of 12 weeks subsequent to the intervention, the participants demonstrated noteworthy enhancements (*p* < 0.001) in their depression and anxiety scores, their overall Mental Component scores in the Medical Outcomes Study Short Form 12, as well as sexual function. Importantly, these gains were sustained at the 24-week juncture post-therapy. (4) Conclusions: The findings stemming from our prospective case study underscore the potential utility of myofascial therapy for women grappling with chronic pelvic pain. This form of intervention yields significant advancements in alleviating anxiety, depression, health-related quality of life, and sexual function.

## 1. Introduction

Chronic pelvic pain (CPP) is characterized by recurrent or persistent discomfort situated in the lower abdominal, pelvic, or intrapelvic regions occurring for a minimum of 3 to 6 months. Notably, this pain in women is not exclusively linked to the menstrual cycle or pregnancy [1,2]. A consensus among studies has emerged regarding the substantial prevalence of this syndrome [3,4], which affects approximately 2–16% of the global populace [5]. The lifetime occurrence of CPP stands at 33%, with women experiencing a notably higher incidence [4].

Epidemiological evidence underscores the heightened vulnerability of women, as compared to men, to chronic pain syndromes [6,7]. Female individuals frequently exhibit a greater propensity for reporting pain in multiple sites, more frequent occurrences of pain, and pain of greater intensity than their male counterparts. Broadly, women dealing with persistent pain typically bear a protracted history of pain, notable psychological distress, and hindrances to both work and physical functioning, and they harbor skepticism toward treatment regimens. These factors collectively contribute to non-compliance with treatment, protracted pain duration, compromised physical and mental well-being, and a decline in overall quality of life. Notably, several authors have advanced the idea of a direct association between CPP and depression [8,9,10] as well as sexual function [11,12,13,14].

Numerous researchers contend that the origin of both discomfort and dysfunction in individuals grappling with CPP resides within the realm of the distressing and persistently tense myofascial tissue surrounding the structures of the pelvic floor [15,16,17,18]. It is pertinent to note that myofascial pelvic pain (MFPP), a subset of this condition, often evades detection and remains untreated within the landscape of CPP [19,20]. The proportion of women affected by CPP who concurrently experience MFPP is estimated to range from 14% to 23% [21].

Nonetheless, a more recent viewpoint has emerged, proposing that CPP is inherently intricate as it has varying constellations of symptoms exhibited by different patients, with each patient responding distinctively to individual treatment approaches. Given the enigmatic nature of pathogenic mechanisms linked to pelvic genitourinary symptoms, a comprehensive understanding of the involvement of painful myofascial tissue remains elusive [22].

MFPP encompasses the experience of discomfort within the pelvic floor musculature and the interconnected fascia. This syndrome can manifest independently, devoid of any concurrent medical condition, or it might emerge as either a precursor or a consequence to urological, gynecological, or colorectal ailments or other issues pertaining to musculoskeletal and neural systems [19,23]. The defining diagnostic hallmark of MFPP within individuals enduring CPP hinges upon the identification of aberrations during examination. This notably includes the presence of myofascial trigger points within the overactive muscles of the pelvic floor, and distantly, muscle tenderness and restrictions within the connective tissues of the pelvic floor, hip girdle, and abdominal wall [15,19,24,25,26,27].

Physical therapy and conservative therapeutic strategies are in accordance with established treatments for CPP, encompassing pharmacological and injection therapies [1]. Electrotherapy focuses on pain modulation [28], while manual therapy involves myofascial trigger point release, forming the foundation for myofascial therapy (MT). MT serves as a specific therapeutic approach aimed at addressing the symptoms and clinical presentation of CPP [29]. Promising evidence has surfaced, suggesting the capacity to alleviate CPP symptoms through the implementation of myofascial physical therapy techniques [17,25,27].

Consequently, the primary aim of this investigation was to evaluate the extent of anxiety, depression, health-related quality of life (HR-QoL), and sexual dysfunction within women afflicted by CPP, and subsequently, to elucidate the outcomes ensuing from the application of MT in these particular patients.

## 2. Materials and Methods

### 2.1. Design

This research adopted a prospective case series approach aiming to comprehensively assess anxiety, depression, HR-QoL, and sexual dysfunction among women grappling with CPP. This study spanned from November 2022 to May 2023 and involved the recruitment of a total of 23 female participants. The selection process involved identifying women seeking physical therapy across designated clinical sites while strictly adhering to the criteria for inclusion and exclusion, as delineated in Appendix A. Eligible participants were adult women who held a confirmed clinical diagnosis of CPP along with co-occurring MFPP. To maintain homogeneity in the dataset due to the protocol’s session requirement, entry was limited to subjects whose symptoms had manifested for a duration of fewer than 5 years. Notably, individuals who had previously undergone myofascial physical therapy for their symptoms were excluded from participation. The enrollment process involved securing written informed consent from each participant before their inclusion. Throughout the study, the protocol adhered to the Ethical Principles for Medical Research in Humans, as laid out in the Declaration of Helsinki, and was approved by the Ethics Committee of the University of Malaga with reference number CEUMA:36-2020-H.

### 2.2. Participants

As part of the standard evaluation procedure for pelvic pain, every enrolled participant underwent a comprehensive assessment to identify myofascial trigger points and anomalies within the pelvic floor and lumbopelvic muscles. During this initial examination, participants were evaluated for tenderness within designated areas. Those individuals in whom tenderness was detected in any of these specified regions during the baseline pelvic examination were deemed eligible to proceed with the study. Conversely, participants lacking such tenderness were deemed ineligible for further participation.

During the second study visit, participants underwent an extensive evaluation of their musculoskeletal system and soft tissues, administered by a skilled physical therapist. For continuation within the study, participants required confirmation from the therapist that tenderness or restriction was indeed present during the pelvic examination. It is noteworthy that the location of detected tenderness did not necessarily need to align with the sites identified by the primary investigator. Furthermore, the physical therapist conducted additional pre-treatment assessments encompassing the mapping of scars and connective tissue restrictions. A comprehensive evaluation of all soft tissues spanning the back, hip girdle, abdominal wall, and pelvic floor also formed an integral part of these preliminary assessments.

### 2.3. Outcomes

Comprehensive sociodemographic information encompassing age, race, educational attainment, occupational status, annual family income, parity, and marital status was diligently gathered from all participants.

The Hospital Anxiety and Depression Scale (HADS) was employed [30], comprising two distinct subscales, with each subscale consisting of seven items. Respondents assessed these items using a four-point scale, scoring in the range of 0–3. As a result, the total scores for each subscale spanned from 0 to 21. The interpretation of scores encompassed three ranges: 0–7, representing ‘no case’; 8–10, indicating a ‘possible case’; and 11–21, indicating a ‘probable case of anxiety/depression’. Notably, these threshold values have undergone validation against clinical interviews, demonstrating a sensitivity and specificity of approximately 0.80 [31].

HR-QoL was assessed with the Medical Outcomes Study Short Form 12 (SF-12) [32]. This instrument furnishes two overarching summary scores: the Physical Component Scale and the Mental Component Scale. Each of these summary component scores spans a continuum from 0 to 100. Notably, elevated scores within this range correspond to an improved state of HR-QoL.

For the measurement of female sexual function, the Female Sexual Function Index (FSFI) emerges as a comprehensive self-report tool [33]. Comprising a total of 19 items, this instrument systematically quantifies six distinct domains of female sexual dysfunction: desire (2 items), arousal (4 items), lubrication (4 items), orgasm (3 items), satisfaction (3 items), and pain (3 items). The cumulative summation of these 19 items yields the total FSFI score, encompassing a range from 2 to 36. Notably, elevated scores within this spectrum signify a more favorable state of sexual function. Significantly, a designated FSFI total score of 26.55 has demonstrated utility as the optimal threshold for discerning between women grappling with and without sexual dysfunction [34].

The evaluation of all outcome measures was conducted at three distinct points: at baseline, at the 12-week milestone, and again at the 24-week point following the intervention. To facilitate comprehensive monitoring, patients were engaged via email on a weekly basis by the study coordinator during the intervening treatment period. This interaction involved inquiries concerning any potential adverse events, which were then duly documented throughout the treatment phase of the study.

### 2.4. Intervention

A structured myofascial therapy (MT) protocol encompassing 15 sessions, each lasting 1 h, was meticulously devised. These sessions were systematically distributed over the course of 12 weeks. To ensure meaningful engagement and a substantial treatment effect, participants were required to complete a minimum of 8 training sessions. This benchmark was essential for their inclusion within the post-intervention assessment phase, aligned with the tenets outlined by Anderson [22]. MT was performed by two adept and certified physical therapists who are well versed in the practice. This intervention was centered on the precise targeting of trigger points and taut bands through the application of myofascial release techniques. These therapeutic procedures entail the application of hands-on external and, at times, internal methods tailored to the unique physiology of each specific tissue. For instance, techniques encompassed gentle, gradual, sustained pressure; flat palpation; deep tissue release; connective tissue release; and the mobilization of visceral fascial structures. Additionally, both physical therapists typically offered pertinent postural guidance and recommended home exercises, often involving stretching routines, to each participant.

### 2.5. Statistical Analysis

SPSS software (version 15.0, SPSS Inc., Chicago, IL, USA) was used to conduct statistical analyses. A threshold *p*-value < 0.05 (two-tailed) was indicative of statistical significance. Quantitative variables and shifts relative to the baseline were characterized by employing statistical descriptors encompassing range, mean, standard deviation (SD), median, and interquartile range (IR).

The nature of our study, being a prospective case series, involved the systematic collection and analysis of data from a consecutive series of participants rather than relying on a pre-determined sample size calculation. Given the nature of the case series design, a formal sample size calculation was not deemed applicable. We aimed to enroll all eligible participants during the study period to maximize the richness and diversity of our dataset.

The study methodology incorporated an intention-to-treat analysis strategy, encapsulating all participants within the analytical framework. This approach provides a comprehensive assessment of the efficacy of the intervention. The normality of data distribution was checked through the Shapiro–Wilk test. Against potential type I errors arising from non-normality in certain variables and the relatively modest sample size, the evaluation of intra-subject changes involved a twofold strategy. Parametric tests (*t*-test) and nonparametric methods (range) were employed. This blend of approaches optimizes the robustness of the analysis, given the data’s specific characteristics.

## 3. Results

The study cohort consisted of a total of 18 participants. However, due to various factors, including an inability to complete the protocol for 2 individuals and an inability to follow-up with an additional participant, the final analytical sample was composed of 15 participants (Figure 1). The data were normally distributed. The baseline demographic characteristics of the sample population were characterized by the following: The participants had an average age of 43.3 ± 10.6 years. The majority of participants identified as white/Caucasian (95%) and were employed (55%). Moreover, 35% of the participants were nulliparous, while 90% were either married or engaged in a partnership.

The myofascial therapy (MT) protocol exhibited a commendable compliance rate of 100%, with all participants successfully concluding every scheduled session. Impressively, none of the treated participants reported any adverse effects or harm arising from the intervention. However, two participants were excluded in week 12, which was attributed to incomplete questionnaires. Additionally, a separate instance of discontinuation was noted in week 24, stemming from an inability to establish contact with the respective participant.

The outcome measurements in the study sample are presented in Table 1. Insights were collected by drawing upon established cut-off points for the Hospital Anxiety and Depression Scale (HADS) and the Female Sexual Function Index (FSFI), alongside reference norms from the SF-12 for age-grouped women. Specifically, at baseline, half of the women exhibited indications of anxiety (50%), while a substantial subset manifested signs of depression (40%), with both scores surpassing the respective threshold of 11. Moreover, the baseline SF-12 scores were found to fall beneath the mean reference values: 51.6 (CI: 50.7–52.4) for the Physical Component and 49.3 (CI: 48.3–50.3) for the Mental Component. Lastly, 65% of the participants registered instances of sexual dysfunction, as their scores remained at ≤26.55, aligning with the established parameter.

Figure 2 and Figure 3 illustrate the outcomes for each woman based on the dimensions of anxiety and depression from the HADS and the physical and mental components of the SF-12, respectively, from baseline to weeks 12 and 24 after MT. Figure 4 depicts the outcomes for the groups of women for each dimension of the FSFI questionnaire from baseline to 12 and 24 weeks.

The changes in the outcome measure values, spanning from baseline to the 12-week and 24-week post-treatment intervals, are meticulously detailed in Table 2. Both analytical approaches—parametric and nonparametric—yielded congruent outcomes across the majority of cases. However, a notable exception pertains to the shift seen in the SF-12 Mental Component at the 24-week mark compared to the baseline. Here, the nonparametric test outcomes hovered near significance, while the parametric test results undeniably attained a status of significance.

Employing age as a covariate and categorizing participants into two groups—those below the age of 40 and those above the age of 40—permitted a discerning exploration. When considering these divisions, the majority of measures exhibited a lack of statistically significant disparities between the two groups. Nevertheless, a noteworthy exception emerged within the pain domain of the Female Sexual Function Index (FSFI) at the 12-week juncture. Specifically, the comparison revealed a disparity in the pain domain scores: there was a marginal discrepancy [mean ± SD: 0.1 ± 0.6] among women below 40 years contrasted with [1.4 ± 0.3] among those above 40 years, attaining significance with a *p*-value of 0.039.

## 4. Discussion

The main objective of this study was to conduct a prospective investigation into the efficacy of MT in addressing the multifaceted dimensions of the psychological, physical, and sexual aspects associated with CPP. The outcomes gleaned from this study illustrate the potential of MT as a viable therapeutic option for CPP. Notably, statistical enhancements were consistently evident across various questionnaires, signifying noteworthy improvements, barring a singular exception—the Physical Component Summary score of the SF-12.

CPP is routinely characterized by its multifaceted manifestation. This complex presentation casts a pervasive influence across diverse dimensions, encompassing social interactions, mental well-being, and physical capacity. As a consequence, this condition often gives rise to a confluence of challenges, including depression and anxiety [24,35,36].

When considering mental health, the female demographic emerges as notably susceptible to the onset of depression, surpassing their male counterparts. This divergence is underscored by a lifetime prevalence of 21.3% among women, which is in stark contrast to the prevalence of 12.7% among men. Moreover, women frequently vocalize experiences of emotional distress that are intimately entwined with their pain [37]. The trajectory experienced by these individuals prior to diagnosis often reflects a pattern marked by discouraging encounters with healthcare professionals who downplay their pain as overblown, or even unreal, culminating in substantial delays in receiving a formal diagnosis [35]. Furthermore, women dealing with CPP frequently articulate negative pain-related cognitions, manifesting as intrusive mental imagery [38] and a heightened susceptibility to sensitization [39].

Living with CPP is characterized by uncertainty about the course of the disease and the future in general, with pervading concerns about crucial aspects of a woman’s life, such as HR-QoL and sexuality [40]. Few studies, however, have examined the influence of depressive symptoms in patients with chronic conditions on HR-QoL [41,42]. Women with CPP frequently experience significant delays from symptom onset to diagnosis, and this has been shown to be associated with a reduced HR-QoL [43]. Thus, women experience a reduction in social activities due to pain, fatigue, and the need for toilet access, or they worry about the onset of pain while they are out. They also feel less able to socialize when they are out (due to being preoccupied with worry about their condition), resulting in reduced confidence [44]. On the other hand, it has been suggested that a reduced quality of life might contribute to the neurobiological underpinnings of chronic pain [39].

Our results agree with those of other studies that observed a decrease in HR-QoL scores in women with CPP [39,43], demonstrating that mental health was the area of HR-QoL that was most negatively affected by CPP, whereas the least affected area was physical activities [14]. These results show that depression and anxiety, when associated with CPP, may increase the negative impact on the HR-QoL of these patients.

Sexual dysfunction and sexual pain disorder characterize pelvic pain. According to the observations of several authors [45,46], sexual dysfunction was observed in 67.8–73% of women with CPP. Our findings showed a baseline mean total score on the FSFI of 17.8 ± 1.9, and 65% of the patients had sexual dysfunction according to the cut-off point (26.55) described for differentiating women with and without sexual dysfunction [34].

Like other authors, we agree with the finding of decreased desire and arousal associated with pelvic pain in women [46,47], with baseline scores of 2.8 ± 0.4 and 2.8 ± 0.3, respectively. Our results showed poor scores in the satisfaction dimension (2.8 ± 0.4), followed by pain during sexual activity (3.2 ± 0.4), which appears to be the sexual dysfunction that is the most highly associated with pelvic pain [48].

Single medical management [49,50] and multidisciplinary approach strategies with psychological and cognitive behavioral therapy [49,51] have proven to be effective in sexual function improvement. Our results partially agree with the observations of others concerning the effectiveness of manual therapy [52,53]. Although there were no improvements in numerous components of sexuality, and sexual function was not restored to nonclinical levels at the 3-month follow-up, we observed a significant improvement in the FSFI total score 12 and 24 weeks after treatment (*p* < 0.001).

The multi-symptomatic presentation of CPP has frequently been described. To understand the presentation of the myofascial pelvic floor in patients with chronic pelvic pain and its influence on mental health, physical activity, and sexual function, it becomes necessary to contemplate fundamental concepts such as viscerovisceral and somatovisceral convergence, the hypertonicity of pelvic floor muscles creating visceral symptoms along with somatovisceral convergence, and central sensitization with an expansion of the receptive fields [23].

Regarding myofascial sources, the etiology could be neuromuscular microtrauma that is the result of sustained positions and movement impairments of the lumbopelvic region. The findings of recent research on the architecture of pelvic floor muscles support the concept that pelvic floor muscles have regional and specific functional demands [54]. Thus, it is logical to consider that if there is an impairment in any one of the contributing components of the levator ani muscle function (muscle, nerve, and connective tissue) and/or if the functional demand increases, pelvic floor muscles may develop myofascial pain syndrome [55].

Current conceptualizations of pelvic floor involvement in sexual dysfunction generally implicate pelvic floor hypertonus as the underlying pathology. Pelvic floor muscle abnormalities, most notably hypertonus, characterized by a paradoxical contraction of the puborectalis when attempting to release, are associated with constipation, incomplete bladder emptying, and penetration difficulties and have been demonstrated to be part of an overall response to heightened anxiety [47,56].

While our study contributes valuable insights into the use and efficacy of manual techniques in CPP, it is essential to acknowledge its inherent limitations. Notably, our study lacks a control group, limiting our ability to draw definitive conclusions about the specific effects of the manual therapy intervention. Additionally, the relatively small number of participants in our study may impact the generalizability of our findings to a broader population. Despite the wealth of literature supporting the use of manual techniques in CPP [15,27,57,58,59,60,61], it is crucial to recognize that the field lacks a consensus on research design. The absence of homogeneity in intervention protocols and the frequency of treatment administration, coupled with the absence of standard treatment protocols guiding manual therapy, introduces variability that may influence the interpretation of our results. These limitations underscore the need for further research with rigorous study designs, larger sample sizes, and the inclusion of control groups to enhance the robustness and generalizability of findings in this complex domain.

Physical therapy must be initiated early in the course of the disease by therapists who are trained in these recent techniques. Additionally, multidimensional treatment strategies addressing the myofascial component, psychosocial factors, and pain are essential for the successful management of CPP.

The results of uncontrolled studies may overestimate effectiveness in comparison to historical controls, may overlook the effect of patient noncompliance, and may ignore harm resulting from the intervention. Further studies with randomized controlled trials comparing different treatment techniques for CPP are needed.

The present study stands fortified by its rigorous exploration into the multifaceted landscape of CPP among women. By addressing the psychological, physical, and sexual dimensions of CPP, our investigation contributes to a deeper understanding of the condition’s intricate implications.

## 5. Conclusions

MFPP is a major component of CPP that is often not properly identified by health care providers. The results of this prospective case report study suggest that MT can be useful for women with CPP, providing significant improvements in anxiety and depression, HR-QoL, and sexual function.

## Figures and Tables

**Figure 1 healthcare-12-00304-f001:**
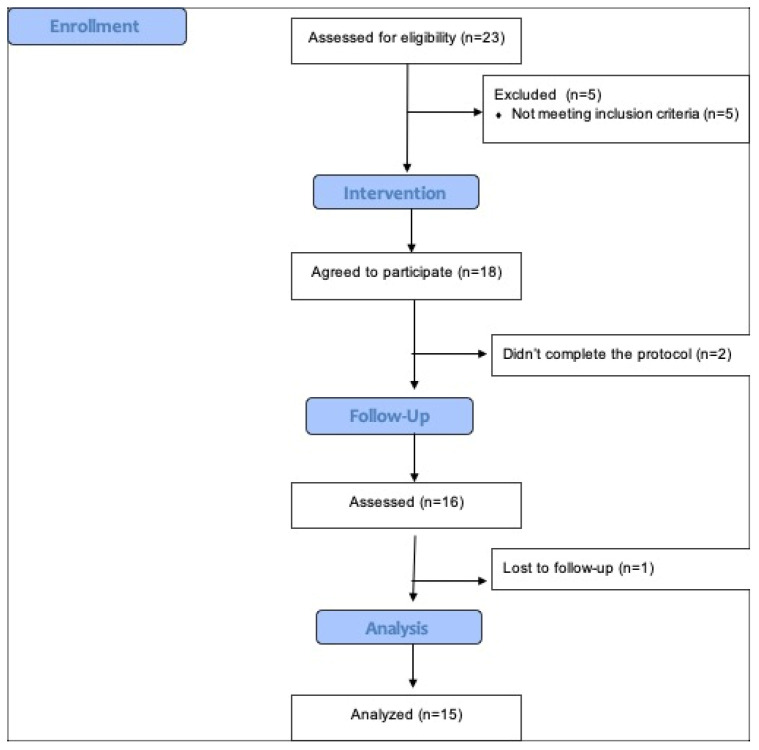
Participant flow diagram.

**Figure 2 healthcare-12-00304-f002:**
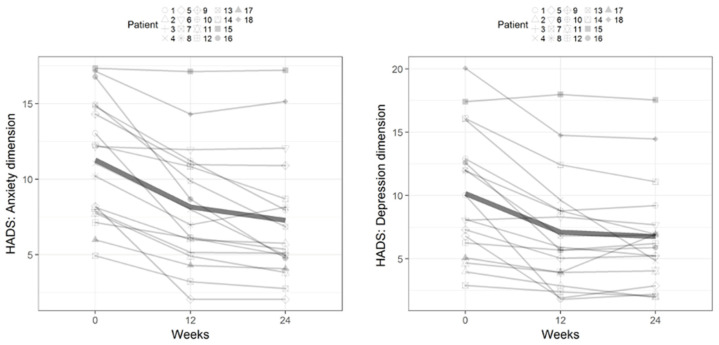
Outcomes for the anxiety and depression dimensions of the Hospital Anxiety and Depression Scale from baseline to 12 and 24 weeks after myofascial therapy.

**Figure 3 healthcare-12-00304-f003:**
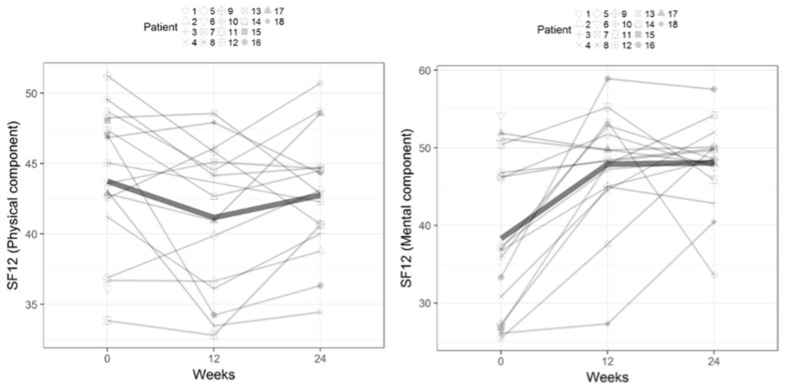
Outcomes for the physical and mental components of the Medical Outcomes Study Short Form 12 from baseline to 12 and 24 weeks after myofascial therapy.

**Figure 4 healthcare-12-00304-f004:**
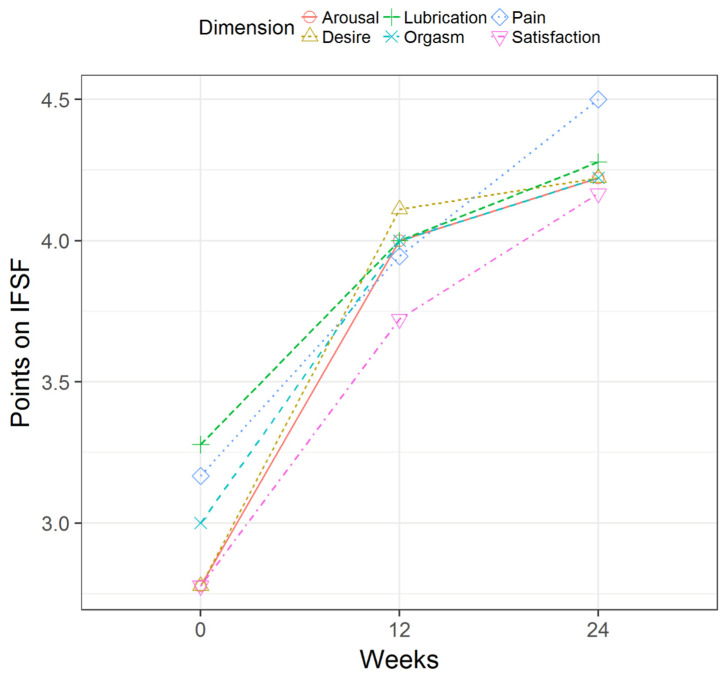
Outcomes for the six dimensions of the Female Sexual Function Index questionnaire from baseline to 12 and 24 weeks after myofascial therapy.

**Table 1 healthcare-12-00304-t001:** Values at baseline for outcome measures in women with chronic pelvic pain.

Measures	Mean ± SD	Range	Median ± IR
HADS: Anxiety dimension	11.3 ± 4.0	5.0–17.0	11.5 ± 6.8
HADS: Depression dimension	10.2 ± 5.1	3.0–20.0	9.0 ± 6.8
SF-12 (Physical component)	43.8 ± 5.0	34.3–51.0	44.0 ± 6.0
SF-12 (Mental component)	38.3 ± 10.1	25.6–54.0	36.7 ± 18.9
FSFI	17.8 ± 7.8	6.0–29.0	17.5 ± 14.6
FSFI Domain: Desire	2.8 ± 1.7	0.0–5.0	2.5 ± 2.8
FSFI Domain: Arousal	2.8 ± 1.4	0.0–5.0	3.0 ± 1.8
FSFI Domain: Lubrication	3.3 ± 1.2	1.0–5.0	3.0 ± 1.8
FSFI Domain: Orgasm	3.0 ± 1.6	0.0–5.0	3.0 ± 2.0
FSFI Domain: Satisfaction	2.8 ± 1.5	0.0–5.0	3.0 ± 2.0
FSFI Domain: Pain	3.2 ± 1.7	0.0–6.0	3.0 ± 2.0

FSFI: Female Sexual Function Index; HADS: Hospital Anxiety and Depression Scale; IR: Interquartile Range; SD: Standard Deviation; SF-12: Medical Outcomes Study Short Form 12.

**Table 2 healthcare-12-00304-t002:** Changes in outcome measure values from baseline to 12 and 24 weeks after MT.

Type of Analysis		Parametric Analysis		Non-Parametric Analysis		
Measures	Time	Mean ± SD	*p*	Median ± IR	*p*	Range
HADS: Anxiety dimension	12 weeks	−3.1 ± 2.2	<0.001 *	−3.0 ± 1.8	0.001 *	−8.0–0.0
	24 weeks	−4.0 ± 3.2	<0.001 *	−3.0 ± 4.0	0.001 *	−12.0–0.0
HADS: Depression dimension	12 weeks	−3.1 ± 2.5	<0.001 *	−2.5 ± 4.0	0.008 *	−8.0–0.0
	24 weeks	−3.4 ± 3.3	<0.001 *	−3.5 ± 4.0	0.031 *	−11.0–2.0
SF-12 (Physical component)	12 weeks	−2.0 ± 4.5	0.090	−0.7 ± 6.1	0.815	−12.4–4.0
	24 weeks	−1.0 ± 5.6	0.488	−0.7 ± 5.8	0.815	−10.5–8.3
SF-12 (Mental component)	12 weeks	8.0 ± 9.3	0.003 *	6.9 ± 15.0	0.008 *	−9.7–25.3
	24 weeks	8.8 ± 10.9	0.004 *	5.0 ± 19.5	0.031 *	−4.4–27.3
FSFI	12 weeks	5.4 ± 4.7	<0.001 *	4.0 ± 2.0	<0.001 *	−2.0–20.0
	24 weeks	7.8 ± 6.4	<0.001 *	6.5 ± 6.6	<0.001 *	−2.0–21.0
FSFI Dimension: Desire	12 weeks	1.3 ± 1.1	<0.001 *	1.5 ± 1.8	<0.001 *	0.0–4.0
	24 weeks	1.4 ± 1.4	<0.001 *	2.0 ± 2.0	<0.001 *	−1.0–4.0
FSFI Dimension: Arousal	12 weeks	1.2 ± 1.4	0.002 *	1.0 ± 1.8	<0.001 *	0.0–5.0
	24 weeks	1.4 ± 1.2	<0.001 *	1.0 ± 1.0	<0.001 *	0.0–5.0
FSFI Dimension: Lubrication	12 weeks	0.7 ± 1.0	0.009 *	0.5 ± 1.0	<0.001 *	−1.0–3.0
	24 weeks	1.0 ± 1.3	0.006 *	1.0 ± 2.0	0.001 *	−2.0–3.0
FSFI Dimension: Orgasm	12 weeks	1.0 ± 1.0	<0.001 *	1.0 ± 1.6	<0.001 *	−1.0–3.0
	24 weeks	1.2 ± 1.4	0.002 *	1.0 ± 1.0	<0.001 *	0.0–4.0
FSFI Dimension: Satisfaction	12 weeks	0.9 ± 1.3	0.008 *	1.0 ± 1.0	<0.001 *	−1.0–4.0
	24 weeks	1.4 ± 1.5	0.001 *	1.0 ± 2.0	<0.001 *	0.0–5.0
FSFI Dimension: Pain	12 weeks	0.8 ± 1.4	0.035 *	1.0 ± 2.0	0.001 *	−2.0–3.0
	24 weeks	1.3 ± 1.7	0.005 *	1.0 ± 2.0	<0.001 *	−1.0–5.0

FSFI: Female Sexual Function Index; HADS: Hospital Anxiety and Depression Scale; IR: Interquartile Range; SD: Standard Deviation; SF-12: Medical Outcomes Study Short Form 12. * *p* < 0.05.

## Data Availability

The data that support the findings of this study are available upon reasonable request to the corresponding author.

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
