# Peer review of "Using Myofascial Therapy to Improve Psychological Outcomes, Quality of Life, and Sexual Function in Women with Chronic Pelvic Pain—A Case Series"

_healthcare, 2024, doi:10.3390/healthcare12030304_

Round 1

Reviewer 1 Report

Comments and Suggestions for Authors

Dear authors, congratulations for your hard work. It is true, that Chronic Pelvic Pain and its treatment are the Holy Grail for many physicians, since till now we don't have a standard treatment for these patients. It would be a neglection not to mention that many advances have been made towards that direction, but still the pathophysiology, let alone the way of managing these patients is still controversial and unclear. 

Having that said, lets focus on your manuscript. If I may, i would like to share some thoughts with you, that I hope you will find helpful.

1. I wonder what treatment the patients you have enrolled in your study has been given to them, prior the myofascial therapy.

2. You do not have a control group. As you know, having a control group is mandatory in every research. 

3. The number of the participants is small, although the time of follow-up seems adequate.

4. You do not mention any limitations regarding your study. Acknowledging your limitations gives strength, at least in my eyes, to your study.

Besides that, I strongly believe that you are in the right tract. CPP deserves a multimodal approach, and physiotherapy, although has a strong level of recommendation in EAU (European Urology Association) guidelines, has to be a part of our armamentarum when dealing with such patients.

Author Response

REVIEWER 1

Dear authors, congratulations for your hard work. It is true, that Chronic Pelvic Pain and its treatment are the Holy Grail for many physicians, since till now we don't have a standard treatment for these patients. It would be a neglection not to mention that many advances have been made towards that direction, but still the pathophysiology, let alone the way of managing these patients is still controversial and unclear. 

Having that said, lets focus on your manuscript. If I may, i would like to share some thoughts with you, that I hope you will find helpful.

COMMENT 1. I wonder what treatment the patients you have enrolled in your study has been given to them, prior the myofascial therapy.

RESPONSE 1: The authors are grateful for the Reviewer's constructive comments and would like to express their thanks for the valuable feedback. Before enrollment, patients underwent pharmacological treatment without concurrent physical therapy during the intervention.

COMMENT 2. You do not have a control group. As you know, having a control group is mandatory in every research. 

RESPONSE 2: Authors appreciate the insightful comment regarding the absence of a control group. We acknowledge the importance of including a control group in research studies, and we apologize for the oversight in this regard. We have discussed potential implications on the interpretation of our Results in the Discussion section. We understand the critical role of control groups in research design and will ensure their inclusion in future studies.

COMMENT 3. The number of the participants is small, although the time of follow-up seems adequate.

RESPONSE 3: Authors would like to thank the Reviewer for noting the concern regarding the small number of participants in our study. We appreciate your acknowledgment of the adequate follow-up time. We recognize the limitation posed by the sample size and are actively exploring opportunities to address this in future research, as recruiting patients with this condition poses a challenging limitation.

COMMENT 4. You do not mention any limitations regarding your study. Acknowledging your limitations gives strength, at least in my eyes, to your study.

RESPONSE 4: Authors appreciate the keen observation regarding the absence of explicit mention of limitations in our manuscript. We want to acknowledge that while limitations were considered, we failed to clearly state them in the respective section. We recognize the importance of transparently addressing the limitations of our study, as it contributes to the overall strength and credibility of the research. We have revised the manuscript to ensure that limitations are explicitly discussed at the end of the Discussion section. We apologize for any confusion caused by this oversight and sincerely appreciate your guidance in enhancing the clarity and completeness of our work.

Besides that, I strongly believe that you are in the right tract. CPP deserves a multimodal approach, and physiotherapy, although has a strong level of recommendation in EAU (European Urology Association) guidelines, has to be a part of our armamentarum when dealing with such patients.

RESPONSE: We sincerely appreciate your positive feedback and endorsement of our approach in addressing chronic pelvic pain (CPP). Your recognition of the importance of a multimodal approach aligns with our perspective, and we are encouraged by your affirmation.

Reviewer 2 Report

Comments and Suggestions for Authors Considering that Chronic Pelvic Pain represents a prevalent condition that afflicts women, leading to psychological distress and sexual dysfunction, research in this field is current and not negligible. With this study, the authors highlight the potential usefulness of myofascial therapy for these women. According to their findings, this form of intervention makes significant progress in alleviating anxiety, depression, health-related quality of life and sexual function. The manuscript is interesting and well presented; furthermore, the methodology is adequate. Probably the only weak point is the limited number of the study sample, composed of eighteen women suffering from chronic pelvic pain. The suggestion is to increase the density of the study sample. Comments on the Quality of English Language

Minor editing 

Author Response

Authors would like to thank the Reviewer for the feedback. Chronic Pelvic Pain is undeniably a widespread condition that warrants more in-depth investigation for effective patient management. The authors acknowledge the Reviewer's comments and comprehend the concerns regarding the study sample. Ideally, a larger sample size would have been preferable. However, recruiting patients with this condition poses significant challenges, given the numerous barriers that hinder their participation in both research and clinical settings.

Reviewer 3 Report

Comments and Suggestions for Authors

The manuscript by Mohedo et al., Effects of Myofascial Therapy to improve psychological outcomes, quality of life, and sexual function in women with Chronic Pelvic Pain. A case series describes the psychological distress and sexual dysfunction related with chronic pelvic pain (CPP) experienced in women. The authors present evidence to support the application of the manual therapy in CPP management. The manuscript is stepwise structured and the experiment procedures are carefully designed to match the conclusions that are given.

I have carefully read through the manuscript and have some concerns which I believe would help to improve understanding and add clarity to the work.

1.   In the Introduction, the authors should outline more detailed the types and effects of myofascial therapy on CPP patients. The knowledge and review of physiotherapy management of CPP is missing in the introduction part, regarding the background of the most authors.

2. In the Methods part, there is information regarding patients inclusion criteria missing. How about the use of pain-management medications? Were the participants allowed to take medications? How about the primary causes of pain, did you screen the participants thoroughly? Nevertheless authors provide detailed inclusion and exclusion criteria in the supplementary material, this does not reflect in the main paper.

3. In the Methods part, authors mention about home exercise prescription to each participant. Did these programs differ between patients? How did you elucidate the exact effect of manual therapy then, if patients were prescribed physical exercises? Must be clarified.

4. In the statistical analysis part, what was the measure to check the normality of data distribution? The necessity to apply both parametric and non-parametric tests is doubtful.

5. In the Results part, authors provide information about participants' education level and income, however this data seems redundant and does not add any value to the manuscript.

6. In the Results part, Figure 1 (Participant Flow Diagram) is completely not clear and should be redesigned to be more informative and clear (i.e. the time frames of intervention and follow-up).

7. Simple table of participants' baseline characteristics would be useful for reader, instead of plain text.

8. The overall presentation of results is scarce. Written text does not link to the data presented in the tables or is not clearly described. Tables seem to be overcrowded with the data.

9. In the Results part, suddenly appears the categorization of participants by age. It is not clear, what authors expected by this categorization, since this aspect was not covered and referenced in the introduction part.

10. In the Discussion part, the use of numeric values of the study results is not recommended. Paragraph 4, lines 275-276, the mentioned interventions are not related with the myofascial therapy and should not be taken as a reference when discussing effect size of the applied intervention.

11. The limitations of the current study are not described.

Comments on the Quality of English Language

The overall presentation of the manuscript feels somehow unusual, regarding the structure of sentences, the regular words and specific terms used (i.e. 24-week juncture post-therapy; global populace; CPP hinges upon; aberrations; intrusive mental imagery etc.). I would suggest to consult English editor to improve the quality of English Language.

Author Response

REVIEWER 3

The manuscript by Mohedo et al., Effects of Myofascial Therapy to improve psychological outcomes, quality of life, and sexual function in women with Chronic Pelvic Pain. A case series describes the psychological distress and sexual dysfunction related with chronic pelvic pain (CPP) experienced in women. The authors present evidence to support the application of the manual therapy in CPP management. The manuscript is stepwise structured and the experiment procedures are carefully designed to match the conclusions that are given.

I have carefully read through the manuscript and have some concerns which I believe would help to improve understanding and add clarity to the work.

COMMENT 1.  In the Introduction, the authors should outline more detailed the types and effects of myofascial therapy on CPP patients. The knowledge and review of physiotherapy management of CPP is missing in the introduction part, regarding the background of the most authors.

RESPONSE 1: Author’s would like to thank the Reviewer for the valuable and thorough review, understanding the feedback’s intention is to improve our reporting. The Reviewer rightly emphasizes the need for a more detailed exploration of myofascial therapy's types and effects on chronic pelvic pain (CPP) patients in the Introduction. Authors have included a paragraph in the Introduction including this exploration. This addition ensures a comprehensive background, contextualizes the study within broader physiotherapeutic approaches to CPP, and aligns with the reviewer's concern for a more thorough introduction to myofascial therapy's relevance to CPP patients.

COMMENT 2. In the Methods part, there is information regarding patients inclusion criteria missing. How about the use of pain-management medications? Were the participants allowed to take medications? How about the primary causes of pain, did you screen the participants thoroughly? Nevertheless, authors provide detailed inclusion and exclusion criteria in the supplementary material, this does not reflect in the main paper.

RESPONSE 2: Authors would like to thank the Reviewer for the insightful feedback. We appreciate the attention to detail. To address the concern, we want to clarify that the information regarding patient inclusion criteria, including the use of pain-management medications and the screening for primary causes of pain, is indeed provided in detail in the supplementary material, as we wanted the manuscript to be as concise as possible. We would like to deeply apologize for This issue.

COMMENT 3. In the Methods part, authors mention about home exercise prescription to each participant. Did these programs differ between patients? How did you elucidate the exact effect of manual therapy then, if patients were prescribed physical exercises? Must be clarified.

RESPONSE 3: Authors understand the Reviewer’s concern and would like to apologize for the misunderstanding. It's important to note that the home exercise programs were standardized and did not differ between participants intentionally. The uniformity in the home exercise prescription was a deliberate choice aimed at mitigating the potential confounding effect of varied exercise regimens on the evaluation of manual therapy intervention. By keeping the home exercise programs consistent across participants, we sought to isolate and elucidate the specific effects of manual therapy without the influence of divergent exercise interventions.

COMMENT 4. In the statistical analysis part, what was the measure to check the normality of data distribution? The necessity to apply both parametric and non-parametric tests is doubtful.

RESPONSE 4: Authors would like to thank the Reviewer for the valuable observation. In our statistical analysis, we employed the Shapiro-Wilk test to assess the normality of data distribution, and we have incorporated a statement accordingly. Authors also appreciate the concern about the necessity of both parametric and non-parametric tests. Our decision to include both types of tests was driven by the potential impact of outliers or deviations from normality on the robustness of our results.

COMMENT 5. In the Results part, authors provide information about participants' education level and income, however this data seems redundant and does not add any value to the manuscript.

RESPONSE 5: Authors acknowledge the Reviewer’s concern regarding perceived lack of value in presenting this data. To address this, we carefully reevaluated the relevance of including participants' education level and income in the context of our Study, and we opted to omit this information, as we align with the Reviewer’s opinion.

COMMENT 6. In the Results part, Figure 1 (Participant Flow Diagram) is completely not clear and should be redesigned to be more informative and clear (i.e. the time frames of intervention and follow-up).

RESPONSE 6: Authors acknowledge the concern regarding the clarity of Figure 1, the Participant Flow Diagram. The design of the flow chart follows the CONSORT guidelines template, adhering to the specifications of the journal for reporting clinical trials. While we understand the desire for increased clarity, the use of the CONSORT template is intentional, aligning with widely accepted standards for transparent reporting of trial procedures. The utilization of the CONSORT template serves a purpose in promoting consistency and adherence to reporting guidelines, ensuring the accurate and transparent depiction of participant flow throughout the study.

COMMENT 7. Simple table of participants' baseline characteristics would be useful for reader, instead of plain text.

RESPONSE 7: Authors would like to thank the Reviewer for the suggestion. We appreciate the interest in enhancing the presentation of participants' baseline characteristics. In our study, given the absence of multiple groups for direct comparison, and having a singular intervention group, we opted for a detailed textual description of baseline characteristics to provide a comprehensive overview of the sample. We believe the information conveyed in the text is sufficient for readers to grasp the essential demographic details. However, we again would like to thank the Reviewer for the suggestion.

COMMENT 8. The overall presentation of results is scarce. Written text does not link to the data presented in the tables or is not clearly described. Tables seem to be overcrowded with the data.

RESPONSE 8: Authors acknowledge the Reviewer’s concern and would like to thank the Reviewer for the valuable feedback. We appreciate the comments on the presentation of results. While we understand the concern about the clarity of the text-to-table connection, the detailed information provided in the text is intended to complement the tables and offer a comprehensive understanding of the study outcomes. While we acknowledge the Reviewer’s concern about potential overcrowding, this intentional use of detailed text is intended to enhance the clarity and depth of information presented. Our commitment is to maintain the integrity of the manuscript while optimizing the overall presentation to ensure reader comprehension. We appreciate the Reviewer’s valuable input, and our goal is to enhance the coherence between the narrative and tables without necessitating significant modifications.

COMMENT 9. In the Results part, suddenly appears the categorization of participants by age. It is not clear, what authors expected by this categorization since this aspect was not covered and referenced in the introduction part.

RESPONSE 9: Authors appreciate the Reviewer’s comment and understand the concern. The categorization of participants by age in the Results section was introduced to explore potential age-related trends or variations in the study outcomes. While this aspect may not have been explicitly referenced in the introduction, it was a post hoc analysis conducted to investigate any noteworthy patterns that emerged during data analysis. Despite not being pre-specified, this additional exploration is valuable for identifying potential age-related nuances in the study results. Although not explicitly detailed in the introduction, the age categorization was undertaken in response to observed patterns during the course of data analysis, aiming to provide a more comprehensive understanding of the study findings.

COMMENT 10. In the Discussion part, the use of numeric values of the study results is not recommended. Paragraph 4, lines 275-276, the mentioned interventions are not related with the myofascial therapy and should not be taken as a reference when discussing effect size of the applied intervention.

RESPONSE 10: Authors value the constructive comment. We concur with the recommendation to avoid the use of numeric values in discussing study results in the Discussion section. Specifically, we acknowledge that the interventions mentioned in Paragraph 4, lines 275-276, are not directly related to myofascial therapy and should not serve as a reference point for discussing the effect size of the applied intervention. To address this concern, we have opted to eliminate the paragraph in question. This decision aligns with the Reviewer's feedback and aims to enhance the precision and relevance of the discussion.

COMMENT 11. The limitations of the current study are not described.

RESPONSE 11: Authors would like to thank the Reviewer for the valuable feedback. While the limitations were not explicitly outlined as a separate section, we have incorporated them at the end of the Discussion section. We recognize the importance of clearly delineating limitations to enhance transparency and the critical evaluation of our findings. To address this concern, we have added a sentence to explicitly indicate the section where the limitations are presented.

Round 2

Reviewer 1 Report

Comments and Suggestions for Authors

Dear authors, although you acknowledge my remarks you haven't embeded your answers  in your paper.

1. you do not mention the kind of pharmaceutic therapy which patients had received prior the study

2. you have added only a phrase and I quote "Our study is not exempt of limitations" without mentioning and exploring your limitations such as control group missing, small number of participants etc.

Author Response

REVIEWER 1 

Dear authors, although you acknowledge my remarks you haven't embedded your answers in your paper.

COMMENT 1. you do not mention the kind of pharmaceutic therapy which patients had received prior the study.

RESPONSE 1: Authors would like to thank the Reviewer for the continued engagement with our manuscript. Prior to the myofascial therapy intervention, patients in our study underwent pharmacological treatment, primarily consisting of analgesic medication. We hope this addresses your query, and we remain open to any further suggestions or inquiries.

COMMENT 2. you have added only a phrase and I quote "Our study is not exempt of limitations" without mentioning and exploring your limitations such as control group missing, small number of participants etc.

RESPONSE 2: Authors would like to thank the Reviewer for the feedback. We have addressed the oversight in the limitations section by including details on the absence of a control group and the small participant size (lines 337-349). Your input has been invaluable, and we appreciate your time and consideration.

Reviewer 3 Report

Comments and Suggestions for Authors

The authors revised and improved the manuscript.

In the added paragraph in the introduction (lines 73-79) there is no need to explain CPP abbreviation, since it was already mentioned at the beginning of introduction.

In the same added paragraph, it is not clear what authors mean by "MT represents a targeted methodology for addressing these abnormalities" (line 77).

In the statistical analysis part, authors included the information about measure to check normality of data distribution. Unfortunately, it is not indicated, if all the data were normally distributed or not.

I may have missed this in the first review, but how was the sample size calculated?

In Figure 1, the Participant Flow Diagram, there is English language mistake left: "Didn't complete de protocol". Please correct.

Comments on the Quality of English Language

English language has not been improved after first review.

Author Response

REVIEWER 3
The authors revised and improved the manuscript.

COMMENT 1. In the added paragraph in the introduction (lines 73-79) there is no need to explain CPP abbreviation since it was already mentioned at the beginning of introduction.

RESPONSE 1: Authors would like to thank the Reviewer for the thorough review of our manuscript. We appreciate the keen observation regarding the redundancy in explaining the CPP abbreviation in the added paragraph of the introduction. We apologize for the oversight and we addressed this in the revised version. The feedback is invaluable, and we are grateful for the time and constructive input.

COMMENT 2. In the same added paragraph, it is not clear what authors mean by "MT represents a targeted methodology for addressing these abnormalities" (lines 77-78).

RESPONSE 2: Authors appreciate the Reviewer’s careful consideration. We understand the confusion in that statement. In response to the insightful feedback, we have modified this sentence to enhance clarity. 

COMMENT 3. In the statistical analysis part, authors included the information about measure to check normality of data distribution. Unfortunately, it is not indicated, if all the data were normally distributed or not.

RESPONSE 3: Authors would like to thank again the careful review of our manuscript. We have addressed the concern by including the relevant information on data normality in line 189. We apologize for any oversight and appreciate the Reviewer’s diligence in ensuring the completeness of our statistical reporting.

COMMENT 4. I may have missed this in the first review, but how was the sample size calculated?

RESPONSE 4: Authors would like to thank again the Reviewer for the insightful comments and thorough review of our manuscript. We appreciate the diligence in ensuring clarity and completeness. Regarding the Reviewer’s query on the sample size calculation, we have incorporated a dedicated paragraph addressing this aspect in our manuscript (lines 178-182). We apologize for any oversight in not presenting this information earlier and appreciate the Reviewer’s commitment to enhancing the rigor of our study.

COMMENT 5. In Figure 1, the Participant Flow Diagram, there is English language mistake left: "Didn't complete de protocol". Please correct.

RESPONSE 5: Authors thank the Reviewer for catching the language mistake in the Participant Flow Diagram (Figure 1) regarding "Didn't complete de protocol." We appreciate the attention to detail. The error has been duly corrected, and the revised figure now accurately reflects the completion of the protocol. The Reviewer’s meticulous review has contributed to the enhancement of the manuscript.